# Reflectance Confocal Microscopy and Dermoscopy of Facial Pigmented and Non-Pigmented Actinic Keratosis Features before and after Photodynamic Therapy Treatment

**DOI:** 10.3390/cancers15235598

**Published:** 2023-11-27

**Authors:** Ewelina Mazur, Dominika Kwiatkowska, Adam Reich

**Affiliations:** 1Department of Dermatology, Institute of Medical Sciences, Medical College of Rzeszow University, 35-959 Rzeszow, Polanddokwiatkowska@ur.edu.pl (D.K.); 2Doctoral School, University of Rzeszow, 35-310 Rzeszow, Poland

**Keywords:** reflectance confocal microscopy, RCM, dermoscopy, actinic keratosis, AK, photodynamic therapy, PDT

## Abstract

**Simple Summary:**

Actinic keratosis (AK) is a very prevalent precancerous condition that can progress to an invasive form of squamous cell carcinoma. Photodynamic therapy (PDT) is a relatively new therapeutic method, and it has become widely used to treat skin cancer of non-melanoma type, particularly because of its good cosmetic outcome. New noninvasive imaging techniques, such as in vivo reflectance confocal microscopy (RCM) or (video)dermoscopy, can help practitioners make a diagnosis and assess the efficacy of the ongoing treatment. Nevertheless, there are no studies with in vivo RCM assessing the effect of PDT on characteristic features of different AK subtypes. Our objective was to evaluate the prevalence of specific (video)dermoscopy and RCM features of pigmented and classical subtypes of AK before and after photodynamic therapy (PDT) treatment. New noninvasive imaging techniques such as RCM and (video)dermoscopy can help practitioners better visualize the efficacy of the ongoing PDT treatment in either classical or pigmented AKs subtypes.

**Abstract:**

Actinic keratosis (AK), due to its widespread prevalence, as well as the possibility of progression to an invasive form of squamous cell carcinoma, requires treatment regardless of the clinical stage. New imaging techniques, such as in vivo reflectance confocal microscopy (RCM), significantly increase the accuracy of diagnosis and allow noninvasive evaluation of the therapeutic efficacy of the ongoing treatment. Our objective was to evaluate the prevalence of specific (video)dermoscopy and RCM features of pigmented and classical subtypes of AK before and after photodynamic therapy (PDT) treatment. We included patients with facial grade II AKs (25 pigmented, 275 non-pigmented) were included in the study. Skin lesions were evaluated by (video)dermoscopy and RCM at the baseline and three months after PDT. In classic AK, the most frequent dermoscopic findings were fine wavy vessels (96%), scale (92%), microerosions (48%), and “strawberry” pattern (36%), while pigmented AK was characterized mostly by “rhomboidal pattern” (80%), scale (60%), white globules (48%), “jelly sign”, and superficial pigmentation (40%). RCM’s most characteristic classic AK findings were abnormal honeycomb pattern in the spinous layer, epidermal inflammatory infiltrate, and solar elastosis that were present in 96% of lesions. Pigmented AKs presented mostly with dark central areas of parakeratosis (72%), mottled pigmentation (72%), dermal inflammatory infiltrate (64%), solar elastosis (60%), and abnormal honeycomb pattern in the spinous layer (56%). Dermoscopically, PDT resulted in complete disappearance of the “rhomboidal pattern” in both classical and pigmented AKs, “starburst pattern” and “jelly sign” in classical AKs, and inner gray halo, “rosette sign” and central crust in pigmented AKs. Three months after one PDT session, RCM evaluation showed mostly solar elastosis in both classical and pigmented AK subtypes, epidermal inflammatory infiltrate in classical AKs, and dermal inflammatory infiltrate in pigmented AKs. New noninvasive imaging techniques such as RCM and (video)dermoscopy can help practitioners better visualize the efficacy of the ongoing PDT treatment in either classical or pigmented AK subtypes.

## 1. Introduction

Actinic keratosis (AK), also known as solar keratosis, is a very commonly diagnosed skin pathology among Europeans [1,2]. Sun-exposed areas are prone to be affected by AK lesions, especially on the face, scalp, or dorsum of the hands [3]. Cumulative, lifelong ultraviolet (UV) exposure increases the risk of developing AK. Fair-skinned (Fitzpatrick I or II), older individuals are the most commonly affected [1,2]. This association is further strengthened because AK is considered a precursor to cutaneous squamous cell carcinoma (SCC) [4]. AK and SCC lesions share similar genetic abnormalities in, among others, UVB-related mutations in p53 [5]. The basal layer of the epidermis seems to be particularly affected by the UVB rays, as it damages its DNA and catalyzes the SCC formation [6]. UVA radiation, while deeper-penetrating into the skin (dermis), promotes injury through the free (hydroxyl) radicals formation, but contrary to UVB, it is not associated with keratinocyte dysplasia [7]. Apart from UVR-mediated oncogenic properties, chronic sun exposure demonstrated the ability to create a state of epidermal immunosuppression [8].

AKs present several clinical variants, such as hyperkeratotic AK, atrophic, pigmented, lichenoid AK, cutaneous horn, and actinic cheilitis [9]. Based on histopathological examination, it is also possible to categorize AKs into seven subtypes: hypertrophic, atrophic, bowenoid, acantholytic, epidermolytic, lichenoid, and pigmented, that may overlap in a single lesion [10,11]. On the other hand, the most simple classification, easy to use in a clinical setting, is the one that differentiates between non-pigmented and pigmented lesions [12,13]. Different therapeutic strategies have been implemented regarding different AK subtypes [14,15]. Photodynamic therapy (PDT) is an ideal treatment option for AKs because it is a field-directed treatment that targets the field of cancerization. It destructs selectively diseased tissue and is characterized by a good cosmesis, and limited downtime [16]. In vivo reflectance confocal microscopy (RCM) as a noninvasive diagnostic tool that generates characteristics correlating with histopathologic features. In the case of AK, those features encompass atypical keratinocytes, parakeratosis, and vascular alterations. The main RCM features of pigmented AKs are the presence of scaling and parakeratosis, atypical keratinocytes; bright, small, dermal papillae with enlarged interpapillary space; increased epidermal thickness and intraepidermal dendritic cells referable to Langerhans cells [17]. Recently, RCM has been more commonly used for monitoring treatments of AK, including PDT, with good accuracy [18,19]. Our team published an article regarding the PDT treatment of facial pigmented AKs [15]. Nevertheless, there are no studies with in vivo RCM assessing the effect of PDT on characteristic features of different AK subtypes.

## 2. Materials and Methods

### 2.1. Patients

The study was performed in accordance with the Helsinki Declaration of 1964 and its later amendments. Informed consent was obtained from the patients for participation in the study. The study enrolled 52 patients (34 women, 18 men), Fitzpatrick phototype II, with a total of 300 AKs treated. The mean (±standard deviation) age of the patients was 73.61 ± 10.91 years. Patients presenting with facial AKs grade II were diagnosed and assessed based on typical features via both (video)dermoscopic as well as RCM examination.

### 2.2. Study Design

This is a prospective study of patients with facial AK lesions treated from May 2022 to June 2023 at the Dermatology Clinic of Clinical University Hospital, Rzeszow, Poland. A total of 300 facial skin lesions (25 pigmented AKs, 275 non-pigmented AKs) grade II according to Olsen’s 1991 criteria [20] were assessed with a (video)dermatoscope (Canfield D200EVO; Canfield Scientific GmbH, Bielefeld, Germany) and RCM (VivaScope 1500/3000; MAVIG GmBH, Munich, Germany). Out of the 300 facial lesions, 45.7% were located on the forehead, 33% on cheeks, 17.6% on the nose, and 3.7% in the perioral area. All assessments were performed before (day 0) and 3 months after (day 112) the PDT session. The lesional skin was previously degreased with 70% ethanol and gently debrided with a curette. Later, the 5-ALA 10% gel (Ameluz, Biofrontera, Inc., Wakefield, MA, USA) was applied. We removed the occlusive, light-shielding dressings after 180 min of incubation period and washed off residues of the preparation with mild soap and water. Next, AK lesions were illuminated with light (Treviolux; MEDlight GmbH, Herford, Germany, pulsed red light, 630 nm). Each patient received one PDT treatment with a total dose of light equal to 37 J/cm^2^ for each lesion.

### 2.3. Videodermoscopy

Three hundred skin lesions (25 pigmented and 275 non-pigmented), Olsen’s grade II, were assessed using the (video)dermatoscope (Canfield D200EVO; Canfield Scientific GmbH, Bielefeld, Germany) at 20- to 70-fold magnification. First, the assessment was performed without immersion fluid (“dry dermoscopy”), then with the use of the ultrasound gel (“wet dermoscopy”). The presence and prevalence of specific dermoscopic AK criteria were assessed. For the pigmented AK lesions following, gray granularity/globules/dots, gray-brown pseudo-network, structureless brown pigmentation, annular granular pattern, hyperpigmented rim of the follicular opening, asymmetrical pigmented follicular openings, as well as jelly sign were assessed. For the classic AK lesions, features such as red pseudo-network, superficial broken-up network, dotted vessels, the “star-like” appearance at the periphery of the lesion, rhomboidal pattern, scale and crust, the double white clods, rosettes, “strawberry sign”, white structureless clods, white globules, microerosions, fine wavy vessels, white and wide follicular openings were assessed at day 0 and at day 112 (three months) after the PDT session.

### 2.4. Reflectance Confocal Microscopy

All analyzed lesions (275 non-pigmented, 25 pigmented) underwent imaging with a near-infrared (diode laser at a wavelength of 830 nm) reflectance confocal microscope (VivaScope 1500/3000; MAVIG GmBH, Munich, Germany). Per each lesion, a minimum of three mosaics (16 × 16 images of 500 × 500 µm) were obtained, at the superficial epidermal layer, dermo–epidermal junction, and papillary dermal level, respectively. In technically challenging areas, a 3000 handheld device was used. In that case, at least 5 vivastacks (25–30 images of 750 × 750 µm) were acquired.

The RCM criteria, that we analyzed in all AKs, included the following features: disruption/individual cells/detached corneocytes, dark central areas of parakeratosis, keratin-filled invagination, corneal pseudocysts, atypical keratinocytes, epidermal and dermal inflammatory infiltrates, disarranged/loss of epidermal pattern, mottled pigmentation, nuclear polarization of spinous layer, intraepidermal dendritic cells, targetoid cells, ringed areas/small and bright papillae or densely packed papillae, presence of cords, polycyclic papillary contours, plump bright cells, vessels traversing dermal papillae (“bottom-hole”), solar elastosis, linear vessels, and increased vasculature.

All clinical, dermoscopic, and RCM examinations were performed in an independent manner by two dermatologists (E.M. and D.K.) with experience in dermoscopy and RCM. Cohen’s kappa coefficient was used to calculate interobserver agreement between the researchers using Statistica^®^ 13.0 Software for Windows Software (Statsoft Polska, Kraków, Poland). Any discrepancies were discussed until a consensus was reached.

## 3. Results

Cohen’s kappa value of interobserver agreement between researcher E.M. and D.K. regarding clinical, dermoscopy, and RCM features of AK before and after PDT treatment was good, equal to 0.78.

### 3.1. Clinical Assessment

All patients benefited from PDT therapy. In the group of classic AK, as well as in the pigmented AK group, 76% and 24% of all patients achieved grade I and 0, respectively, according to Olsen’s criteria (Figure 1).

### 3.2. Videodermoscopy

A total of 300 skin lesions (275 non-pigmented, 25 pigmented) were assessed using the videodermatoscope (Canfield D200EVO; Canfield Scientific GmbH, Bielefeld, Germany). The presence and prevalence of specific dermoscopic pigmented and classic AK criteria were assessed before (day 0) and after (day 112) the PDT session and are summarized in Table 1. To increase the clarity of data presentation, dermoscopic parameters that were not observed in any of the lesions are not included in Table 1.

#### 3.2.1. Classic AK

In classic AK, the most frequent dermoscopic findings were fine wavy vessels (96.4%), scale (92%), “rosette sign” (63.6%), microerosions (48%), “strawberry” pattern (36.4%), white structureless clods (36%), and white globules (31.6%). “Jelly sign”, “rhomboidal pattern”, and “starburst pattern” were observed less frequently, in 12.4%, 8%, and 7.6% of cases, respectively.

Three months after one PDT session, lesions presented mostly with fine wavy vessels (87.6%), scale (51.6%), “rosette sign” (27.6%), and “strawberry pattern” (16%). A number of lesions presenting with microerosions, white structureless clods, and white globules diminished (by 44.4%, 32%, and 23.2%, respectively), while “jelly sign”, “rhomboidal pattern”, and “starburst pattern” disappeared completely.

#### 3.2.2. Pigmented AK

In pigmented AK, the most frequent dermatoscopic findings were “rhomboidal pattern” (80%), scale (60%), white globules (48%), “jelly sign” and superficial pigmentation (40%), inner gray halo (36%), “annular granular pattern” (32%), while white circles, “rosette sign”, double white clods, fingerprinting, central crust, and grayish areas were not as commonly observed (in 28%, 24%, 20%, 16%, 5%, and 4% of lesions, respectively).

Three months after one PDT session, lesions presented mostly with white globules (24%) and an “annular granular pattern” (20%). A number of lesions presenting with scale, “jelly sign” and superficial pigmentation, white circles, double white clods, and fingerprinting diminished (by 56%, 36%, 24%, 16%, and 12%, respectively), while “rhomboidal pattern”, inner gray halo, “rosette sign”, and central crust disappeared completely (Figure 2).

### 3.3. Reflectance Confocal Microscopy

Three hundred AK lesions (275 non-pigmented, 25 pigmented) underwent RCM imaging with a near-infrared reflectance confocal microscope (VivaScope 1500/3000; MAVIG GmBH, Munich, Germany). The median time spent on one AK lesion acquisition was around 120 s for VivaScope 1500 and about half that time for the 3000 device. However, the assessment of a single patient usually lasted about 30 min, including about 5 min for videodermatoscopy and 20–25 min for RCM. Specific RCM pigmented and classic AK criteria, and their prevalence, were assessed before at day 0 and at day 112 after the PDT session. To increase the clarity of data presentation, RCM parameters that were not observed in any of the lesions are not included in this paragraph. Selected features of RCM characteristic of AK are presented in Figure 3.

#### 3.3.1. Classic AK

In classic AK, the most frequent RCM findings were abnormal honeycomb patterns in the spinous layer (96%), as well as epidermal inflammatory infiltrate and solar elastosis that were present in 95.6% of lesions, whereas dark central areas of parakeratosis, disarranged spinous layer, increased vasculature, and dermal inflammatory infiltrate were observed less frequently, namely in 92%, 87.6%, 39.6%, and 35.6% of lesions, respectively (Table 2).

Three months after one PDT session, lesions presented mostly with solar elastosis (92%), epidermal inflammatory infiltrate (40%), and dark central areas of parakeratosis (28.4%). A number of lesions presenting with abnormal honeycomb pattern in the spinous layer, disarranged spinous layer, dermal inflammatory infiltrate, and increased vasculature diminished (by 72.4%, 68%, 31.2%, and 27.6%, respectively), while detached corneocytes, dendritic processes, and round nucleated cells in the spinous layer disappeared completely.

#### 3.3.2. Pigmented AK

In pigmented AK, the most frequent RCM findings were dark central areas of parakeratosis (72%), mottled pigmentation (72%), dermal inflammatory infiltrate (64%), solar elastosis (60%), and abnormal honeycomb pattern in the spinous layer (56%). On the other hand, disarranged spinous layer, epidermal inflammatory infiltrate, nuclear polarization of spinous layer, detached corneocytes, and polycyclic papillary contours were observed less frequently—in 40%, 40%, 36%, 32%, and 32% of cases, respectively.

Three months after one PDT session, lesions presented mostly with solar elastosis (52%), dermal inflammatory infiltrate (32%), increased vasculature (32%), and plump bright cells (32%).

A number of lesions presenting with mottled pigmentation, dark central areas of parakeratosis, abnormal honeycomb pattern in the spinous layer, dendritic processes, and nuclear polarization of the spinous layer diminished (by 56%, 44%, 36%, 32%, and 32%, respectively), while loss of honeycomb pattern was no longer observed in any of the lesions. The only feature that slightly increased in prevalence three months after PDT was plump bright cells (from 28% to 32%) (Table 2).

## 4. Discussion

As early as in 2009, a group of German researchers decided to test the applicability of RCM for noninvasive monitoring of actinic field cancerization. Patients participating in this study applied imiquimod 5% cream in a defined study area three times weekly for a total of four weeks. Serial RCM evaluations were performed at baseline, two weeks after initiation of therapy, and four weeks after the end of treatment. RCM was able to identify subclinical AK by visualization of residual cellular and nuclear atypia within the spinous cell layer. This suggests that RCM may aid in the detection of incomplete lesion clearance, increasing diagnostic accuracy compared to clinical evaluation alone. According to Ulrich et al., the adjunct use of noninvasive optical techniques may increase the therapeutic efficacy of AK treatments [21]. Those findings are in concordance with our results, in which a single session of PDT treated around 75–80% of the studied lesions; however, in the remaining percentage of lesions, we could still observe features characteristic of AK in both dermoscopy and RCM [21]. Histopathological diagnosis has always been the gold standard in dermatological diagnosis of many disease entities. In 2018, Ishioka et al. assessed the RCM accuracy, sensibility, and specificity for AK in 5% 5-fluorouracil treatment, comparing it with the histopathological examination [22]. A total of 50 lesions located on forearms were enrolled. After initial RCM evaluation, topical occlusive treatment with 5% 5-FU cream was prescribed to the affected region (wrist to elbow) every 12 h for a minimum period of four weeks. Another RCM assessment was carried out thirty days after the end of treatment and a biopsy was taken from the studied sites. The concordance of RCM and pathologists’ diagnosis was substantial (κ = 0.637, *p* < 0.001). This study validated RCM as a noninvasive method capable of monitoring AK therapeutic response with efficacy comparable to histological examination [22].

Over the last decade, new therapeutic options for AK have emerged—e.g., Da Silva Sousa et al. in 2019 evaluated the efficacy of daylight PDT (dl-PDT) in facial and scalp AKs using RCM [23]. The examination was conducted in six patients before, one week after, and three months after the dl-PDT session. The presence of the following RCM AK criteria was assessed: scale, nucleated cells in stratum corneum, intraepithelial nucleated cells, atypical honeycomb pattern, dermal inflammatory cells, actinic elastosis. The study demonstrated a normalization of the honeycomb pattern (mean cure rate 74%) and resolution of AKs faster than the clinical evaluation. Interestingly, according to the authors, PDT results were already seen one week after the therapy and have lasted up to the next RCM examination (three months after the treatment) [23].

RCM can be also used to compare different treatment regimens and assess their supremacy. Mota et al. in 2020, used RCM to collate two different cryotherapy protocols for AK [24]. Grade II AKs on the forearms were subjected to a freezing and thawing time of 10 s for one cycle (group A) or two cycles (group B). At baseline and four weeks after treatment, the same dermatologists assessed RCM pictures. Two cycles induced a more marked improvement in parakeratosis and inflammation in the epidermis. On the other hand, there was no statistically significant variation in hyperkeratosis, keratinocyte atypia, dyskeratosis, dermal inflammation, and fibrosis. Therefore, two cycles of cryotherapy, according to RCM, did not generate more side effects (such as fibrosis) while yielding better results [24].

As definitions and assessments of different RCM features of AK may vary among the researchers, several papers were published to unify the topic of nomenclature in RCM keratinocyte atypia examination [25,26,27]. In 2011, Gareau et al. created a computer model that was able to noninvasively evaluate keratinocyte populations as a quantitative morphometric diagnosis in skin cancer detection. It evaluated the cytoarchitectural disorganization and provided objective and quantitative data for dyskeratosis detection and quantification [25]. However, even though automatic identification of epidermal keratinocytes is a promising technique, it is not yet universally available.

Pellacani et al. created a visual representation of different degrees of keratinocyte atypia in AK and correlated them with corresponding histopathological images [26]. They achieved a good assessor correlation between the RCM and histopathology data that demonstrated that trained raters are able to consistently distinguish different levels of cytological atypia. According to the authors, the identification of ‘key’ RCM images (representative lesional horizontal section of the epidermis at the stratum spinosum level) may represent a reference standard for grading keratinocyte atypia through RCM examination. Pellacani et al. created five different grades of atypia, in which raters should focus on assessing irregularity within the honeycomb pattern, keratinocyte contours, their size, shape, and presence of the nucleus [26]. Seyed Jafari et al. on the other hand, to better correspond to the histopathological examination, proposed a classification with only three degrees of atypia, based on irregularities in the honeycomb pattern, architectural disruption, keratinocyte borders, and their morphology [27].

In recent years, an effort has been made to identify reliable AK response criteria. Curiel-Lewandrowski et al. across different AK severity grades, made a comparison of their clinical and RCM responses [18]. The study included twenty patients randomized to receive either PDT or cryotherapy. The assessments (clinical and RCM) were performed in both AK lesions as well as photodamaged skin at baseline, three and six months. They observed significant reductions in many of the RCM criteria. The authors suggested that features such as hyperkeratosis, atypical honeycomb patterns, stratum corneum disruptions, and disarranged epidermal patterns are the most useful criteria when assessing AK response to treatment. They displayed the highest baseline OR and regardless of AK grade or treatment modality were suggestive of significant response to treatment. The authors made the conclusion that RCM may in the future standardize the therapeutic monitoring of AK. However, a uniform RCM measurement protocol of AK treatment response is needed [18].

In our previous study, we found that PDT is an effective treatment modality for facial pAKs in fair-skinned individuals [15]. In dermoscopy, we observed the low resolution of the “annular granular pattern” and the increase in the presence of so-called “grayish areas”. We proposed that this phenomenon can correspond to the Tyndall effect, possibly created by the PDT-related activation of melanophages in the upper layer of the dermis. Further analysis with RCM focused on different AK features and confirmed our suspicion, as we also observed that the treatment with PDT in pAKs leads to the increase in the presence of so-called “plump bright cells” corresponding to melanophages.

## 5. Conclusions and Limitations

The current study demonstrates that the use of new noninvasive imaging techniques such as RCM and videodermoscopy can better visualize the efficacy of the ongoing AK treatment. Moreover, it can facilitate the diagnosis of subclinical lesions. It also suggests that the efficacy of PDT in pAKs is not inferior to non-pigmented AKs. RCM and videodermoscopy are useful tools in evaluating the efficacy of ongoing PDT in the treatment of solar keratosis in both its pigmented and non-pigmented subtypes. The current study limitations include only Caucasian patients (Fitzpatrick phototype II) as a study population and single-center design.

## Figures and Tables

**Figure 1 cancers-15-05598-f001:**
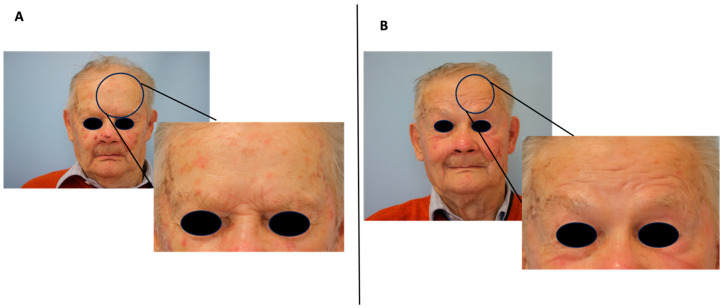
Clinical improvement of classic AK after three months of photodynamic therapy treatment (PDT). (**A**) Before PDT, (**B**) 3 months after PDT.

**Figure 2 cancers-15-05598-f002:**
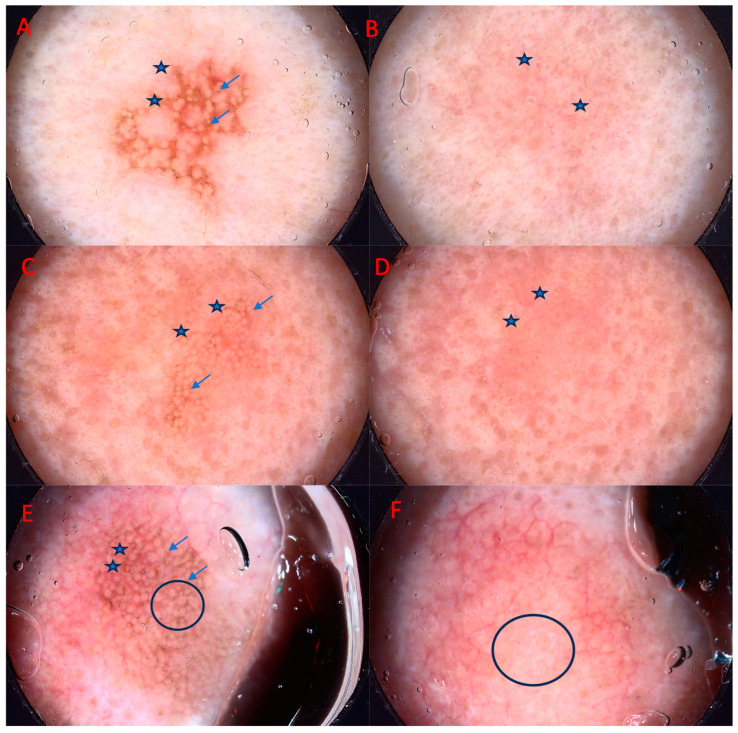
Improvement of pigmented AK after three months of photodynamic therapy treatment (PDT). From left to right: (**A**) asterisk—superficial pigmentation, jelly sign, arrows—rhomboidal pattern, (**B**) astericks—grayish areas, (**C**) asterisk—superficial pigmentation, jelly sign, arrows—rhomboidal pattern, (**D**) asterisk—jelly sign, (**E**) arrows—superficial pigmentation, rhomboidal pattern, asterisk—grayish areas, circle—inner gray halo, (**F**) circle—white globules.

**Figure 3 cancers-15-05598-f003:**
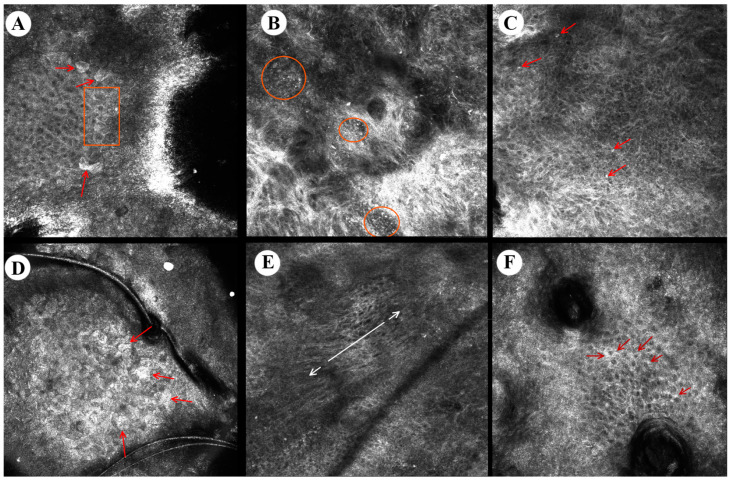
Reflectance confocal microscopy images of actinic keratosis. From left to right: (**A**) detached corneocytes (arrows), parakeratosis (quadrangle), (**B**) dermal inflammatory infiltrate with round lymphocytes (circles), (**C**) epidermal inflammatory round cells (arrows), (**D**) detached corneocytes (arrows), (**E**) nuclear polarization (arrows) in the spinous layer, (**F**) increased intracellular space in stratum granulosum (arrows).

**Table 1 cancers-15-05598-t001:** Change in frequency of dermoscopic classic and pigmented actinic keratosis (AK) criteria at the baseline and three months after single photodynamic therapy treatment (PDT).

Criterion	Classic AK		Pigmented AK	
	% (*n*) at Baseline	% (*n*) Three Months after PDT Treatment	% (*n*) at Baseline	% (*n*) Three Months after PDT Treatment
Fine wavy vessels	96.4 (265)	87.6 (241)	0 (0)	0 (0)
Scale	92 (253)	51.6 (142)	60 (15)	4 (1)
Rosette sign	63.6 (175)	27.6 (76)	24 (6)	0 (0)
Microerosions	48 (132)	3.6 (10)	0 (0)	0 (0)
Strawberry pattern	36.4 (100)	16 (44)	0 (0)	0 (0)
White structureless clods	36 (99)	4 (11)	0 (0)	0 (0)
White globules	31.6 (87)	8.4 (23)	28 (7)	4 (1)
Jelly sign	12.4 (34)	0 (0)	40 (10)	4 (1)
Rhomboidal pattern	8 (22)	0 (0)	80 (20)	0 (0)
Starburst pattern	7.6 (21)	0 (0)	0 (0)	0 (0)
White globules	0 (0)	0 (0)	48 (12)	24 (6)
Inner gray halo	0 (0)	0 (0)	36 (9)	0 (0)
Annular granular pattern	0 (0)	0 (0)	32 (8)	20 (5)
White circles	0 (0)	0 (0)	28 (7)	4 (1)
Double white clods	0 (0)	0 (0)	20 (5)	4 (1)
Fingerprinting	0 (0)	0 (0)	16 (4)	4 (1)
Central crust	0 (0)	0 (0)	4 (1)	0 (0)
Grayish areas	0 (0)	0 (0)	4 (1)	4 (1)

Abbreviations: AK—actinic keratosis, *n*—number of lesions, PDT—photodynamic therapy.

**Table 2 cancers-15-05598-t002:** Change in frequency of reflectance confocal microscopy classic and pigmented actinic keratosis (AK) criteria at the baseline and three months after photodynamic therapy treatment (PDT).

Criterion	Classic AK		Pigmented AK	
	% (*n*) at Baseline	% (*n*) Three Months after PDT Treatment	% (*n*) at Baseline	% (*n*) Three Months after PDT Treatment
Abnormal honeycomb pattern in spinous layer	96 (264)	23.6 (65)	56 (14)	20 (5)
Epidermal inflammatory infiltrate	95.6 (263)	40 (110)	40 (10)	20 (5)
Solar elastosis	95.6 (263)	92 (253)	60 (15)	52 (13)
Dark central areas of parakeratosis	92 (253)	28.4 (77)	72 (18)	28 (7)
Disarranged spinous layer	87.6 (241)	19.6 (54)	40 (10)	16 (4)
Increased vasculature	39.6 (109)	12 (33)	52 (13)	32 (8)
Dermal inflammatory infiltrate	35.6 (98)	4.4 (12)	64 (16)	32 (8)
Nuclear polarization of spinous layer	24.4 (67)	4 (11)	36 (9)	4 (1)
Dendritic processes	24 (66)	0 (0)	44 (11)	12 (3)
Loss of honeycomb pattern	20 (55)	3.6 (10)	8 (2)	0 (0)
Detached corneocytes	16.4 (45)	0 (0)	32 (8)	4 (1)
Round nucleated cells in spinous layer	4.4 (12)	0 (0)	24 (6)	4 (1)
Mottled pigmentation	0 (0)	0 (0)	72 (18)	16 (4)
Polycyclic papillary contours	0 (0)	0 (0)	32 (8)	12 (3)
Plump bright cells	0 (0)	0 (0)	28 (7)	32 (8)
Small and bright papillae	0 (0)	0 (0)	24 (6)	4 (1)

Abbreviations: AK—actinic keratosis, *n*—number of lesions, PDT—photodynamic therapy.

## Data Availability

The datasets generated during and/or analyzed during the current study are available from the corresponding author upon reasonable request.

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
