# Peer review of "Reflectance Confocal Microscopy and Dermoscopy of Facial Pigmented and Non-Pigmented Actinic Keratosis Features before and after Photodynamic Therapy Treatment"

_cancers, 2023, doi:10.3390/cancers15235598_

Round 1
Reviewer 1 Report
Comments and Suggestions for Authors
Overall a good paper that is clearly set out.
The results are not surprising and of interest.
Another future paper may examine the effects of PDT following an application of two sessions of PDT.
Some detail on the sites of the AK cases would enhance the paper.
Author Response
We are grateful to the reviewer for the very supportive comments.
We have provided the following information regarding the lesion localization:
Out of 300 facial lesions 45.7% were located on the forehead, 33% on cheeks, 17.6% on the nose, and 3.7% in the perioral area
Reviewer 2 Report
Comments and Suggestions for Authors
Dear Authors, this is a well written paper, the subject is interesting, in particular the efficacy of PDT on pigmented AK and the possibility to asses it by (video)dermoscopy and RCM.
It would be interesting to know the percentages of accordance/discrepancy of the Authors EM and DK regarding clinical, dermoscopy and RCM features of AK before and after treatment.
It would also be interesting to know the median time the Authors spent to perform RMC for a single AK lesion (acquisition of images and observation of the images), in order to understand the amount of time it would take for such examination in a clinical setting. I imagine it takes a lot more than videodermoscopy.
Author Response
We would like to thank the reviewer for the time spent reviewing our manuscript and for the supportive comments. In the revised version of the manuscript, we have provided Cohen's kappa value of interobserver agreement regarding the RCM assessments. In addition, we have also provided the time needed for the acquisition of a single lesion.